# Utilization of long-acting contraceptive methods and associated factors among female healthcare providers in South Wollo Zone hospitals, Northeast, Ethiopia. A cross-sectional multicenter study

**Aragaw Hamza Yimer** [1]*, **Mehdi Shumiye Seid**[2], **Fasil Walelign**[2], **Yitayish Damtie**[2], **Ahmed Muhye Seid**[3]

**1** Department of Anesthesia, College of Medicine and Health Science, Dire Dawa University, Dire Dawa, Ethiopia, **2** Department of Reproductive and Family Health, School of Public Health, College of Health Science, Wollo University, Dessie, Ethiopia, **3** School of Medicine, College of Medicine and Health Science, Dire Dawa University, Dire Dawa, Ethiopia

* aragaw2010@gmail.com

## Abstract

In Ethiopia Long-acting contraception method utilization was found low (22.7%) among female healthcare providers. However, there is no study has been conducted on the utilization long-acting contraception methods among female healthcare providers in the study area. These studies addressed important variables such as socio-demography and individual factors that might affect the use of long-acting contraceptive methods among female healthcare providers. We assessed the utilization of long-acting contraceptive methods and associated factors among healthcare providers in South Wollo Zone public hospitals, Amhara Region, Ethiopia, in 2021.An institutional-based cross-sectional study was conducted among 354 female healthcare workers in the South Wollo Zone hospitals from March to April, 2021. The participants were selected using a systematic random sampling technique. The data were collected using self-administered questionnaires entered into Epidata version 4.1 and exported to SPSS version 25 for analysis. Bi-variable and multi-variable logistic regression analyses were performed. The adjusted odds ratio (AOR), along with a 95% confidence interval (CI), was estimated to measure the association. The significance level was set at a *P*- value under 0.05. The current utilization of long-acting contraceptive methods among female healthcare providers was found to be 33.6% [95%, CI 29–39]. Discussion with a partner [AOR = 2.277,95% CI, (1.026–5.055)], method shift/switched [AOR = 4.302,95% CI, (2.285–8.102)], knowledge of the respondent [AOR = 1.887,95% CI, (1.020–3.491)], and ever birth [AOR = 15.670,95% CI, (5.065–48.49)] were significant factors toward the utilization of long-acting contraceptive methods. The current utilization of long-acting contraceptive methods was found to be low. Therefore, encouraging partner discussions through a targeted information education communication intervention strategy should be intensified to improve long-acting contraceptive methods utilization.

**Data Availability Statement:** All relevant data are within the paper and its Supporting Information files.

**Funding:** Wollo University supported to conduct this study. The funders had no role in study design, data collection and analysis, decision to publish, or preparation of the manuscript.

**Competing interests:** The authors have declared that no competing interests exist.

# 1. Introduction

## 1.1. Background

Family planning is defined as the ability of individuals or couples to anticipate and attain the desired number of children, as well as the desired spacing and timing of their births [1]. Family planning methods can be classified as natural or modern. Modern family planning can be classified into short and long-acting contraceptive methods; furthermore, these long-acting reversible contraceptive methods (LARCs) are divided into those using intrauterine contraceptive devices (IUCDs) and implants [1–3].

Long-acting contraceptives (LACs) are convenient modern birth control methods that have a low failure rate and are safer and more cost-effective than short-acting contraceptives [4]. LARCs protect against pregnancy for at least 3 years for Implants and for at least 12 years for IUCDs; when removed, the return to fertility is prompt [5,6]. Family planning can decrease maternal death by 20–35%, and it is a human right and key to empowering women, decreasing poverty, promoting female productivity, lowering fertility, and increasing child survival and maternal health [3,7]. Over the past four years, an organized international family planning effort has made great progress in expanding the availability and use of voluntary reproductive health and family planning services [8].

Sub-Saharan Africa (SSA), including Ethiopia, faces serious population and reproductive health challenges, as indicated by higher rates of maternal mortality, total fertility, population growth, and a larger unmet need for family planning [9]. For instance, most maternal and newborn deaths can be prevented with confirmed interventions to certify that each pregnancy is desired using modern contraceptives and that each birth is safe [10]. Moreover, avoiding obstacles to the consumption of contraceptives and the increasing demand for family planning could prevent unintended pregnancies and births, abortions, miscarriages, and maternal and infant deaths each year [11].

The Sustainable Development Goal (SDG) plans to ensure universal access to sexual and reproductive healthcare services, including family planning. This has stepped up the implementation of the Health Sector Transformation Plan (HSTP) [12]. Family planning interventions have been identified as major components to be strengthened to reduce maternal and child mortality and morbidity [1].

Pregnancy and childbirth complications are the leading causes of death in low and middle-income countries, accounting for 99% of global maternal deaths. Evidence suggests that this maternal mortality ratio can be, reduced by more than 25% through family planning interventions [13].

Globally, LARC utilization among reproductive-aged individuals is low, which means that 44% of women of reproductive age do not use long-acting permanent contraceptive methods (LAPCMs), such as IUDs, Implants, or sterilization. Ten percent of married or in-union women do not use female sterilization, and 86% do not use IUDs. However, most contraceptive users in Africa use short-term family planning methods [14]. The utilization of long-acting family planning methods (LAFPMs), especially IUCDs and Implants, is very low in Africa, IUCD and Implant use rates are 4.6% and 1% respectively [15]. Similarly, SSA utilization of LAPMs was very low. Around 25% of SSA couples, and 29% of Ethiopian couples, who want to space or limit births, do not use any form of modern contraception [16].

In Ethiopia, the contraceptive prevalence rate (CPR) has increased from 37% in 2000 to 41.1% in 2019. In contrast, TFR declined from 5.5 children per woman in 2000, to 4.6 children per woman in 2016 [2]. Similarly, in the Amhara region, the Ethiopian Demographic Health Survey (EDHS) 2016 report showed that the utilization of long-acting contraceptive methods (LACMs) is 15.1%, which is low, whereas the utilization of injectable contraceptive methods is

63% for unclear reasons [17]. As for the mini-EDHS 2019, the CPR increases from 41% to 36% in EDHS 2016 [2]. The government planned to increase Implant and IUD use to 33% and 15%, respectively, in the method mix [18]. In terms of the unmet need for family planning, LACMs are more often used for spacing and limiting than short-acting ones; however, the utilization of LAFPs is 11%, which is low [2]. Similarly, the utilization of LACMs among female healthcare workers is also low, (22.7%) [19].

Many factors associated with not using LACMs- include inadequate availability, limited service areas or materials, fear of social dissatisfaction, opposition from partners, religious reasons, fear of side effects, health concerns, and lack of knowledge about contraceptive choices and their uses [10,20]. The reasons for the underutilization of LACMs among female healthcare providers include husbands'/partners' supportive attitudes, the number of children they want to have, the desire to have 0–2 children, attitude, and monthly family income [19]. Not using LACMs can cause 187 million unintended pregnancies, 54 million unplanned births, 112 million induced abortions, 1.2 million infant deaths, and 230,000 maternal deaths, resulting in increased family size, reduced production, and income [21].

The Ethiopia Ministry of Health (EMOH) sets Reproductive Health strategies to strengthen the provision of all Family planning (FP) methods, especially long-acting reversible contraceptives (LARCs), as a key strategy for the reduction of unwanted pregnancies and enabling individuals to meet their desired family size [18,22]. Consequently, to increase the access of FP for households at a community level, a family planning extension package was planned. This package is also a key strategic device to reduce maternal death by spacing or preventing pregnancies that follow too early or too close [23]. In line with Ethiopia's FP 2020 commitments, the Ministry of Health (MoH) developed a health sector transformation plan in 2015, increasing the CPR 42–55%. This would mean reaching an additional 6.2 million women and adolescent girls with family planning services by 2020. Additionally, a federal ministry promised to achieve 40% utilization of the LARC methods by 2020 [18].

Even though, in Ethiopia, there were many studies conducted that assessed the utilization of long-acting and permanent method and associated factors among reproductive-age women. However, as far as our literature search, no study was conducted on LACM utilization among female health providers in the study area. Additionally, since most of the studies were showed correctional and case-control studies among reproductive age group women, these studies failed to include important variable-like residency, work experience, fear of side effect, knowledge of LACMs, source of information, previous use of LACMs, misperception, method preference, time to have the next baby, discussed with a partner, and fear of fertility, Which might affect the use of LACMs in female health providers and limited studies in Ethiopia. Hence, this study assessed the utilization of long-acting contraceptive methods and associated factors among female healthcare providers.

## 2. Objective

### 2.1. General objective

✓ To assess long-acting contraceptive methods utilization and associated factor among female healthcare providers in South Wollo Zone governmental hospitals, 2021.

### 2.2. Specific objectives

✓ To determine the proportion of long-acting contraceptive utilization among female healthcare providers in South Wollo Zone governmental hospitals.

✓ To identify factors associated with long-acting contraceptive utilization among female healthcare providers in South Wollo Zone governmental hospitals.

## 3. Methods and materials

### 3.1. Study design and period

An institutional-based cross-sectional study was conducted from March 22 to April 22, 2021, at selected public hospitals in the South Wollo Zone, Amhara region, Ethiopia among female healthcare providers.

### 3.2. Study setting and population

The study was conducted in the South Wollo Zone Hospital of Amhara region in Ethiopia. This is one of the 14 zones of Amhara region. The capital of the south Wollo zone is Dessie city, which is located 401 Km far from capital city, Addis Ababa and 480Km far from Bahirdar. There are a total of thirteen governmental hospitals at south Wollo zone serving three million populations. Nine of them are primary hospitals, two general hospitals and one specialized hospital. There were 771 female health providers in the study area. Randomly selected female healthcare providers who were working in South Wollo zone hospitals during the study period were used as the study population.

### 3.3. Eligibility criteria

**3.3.1. Inclusion criteria.** All female healthcare providers in the reproductive age.

**3.3.2. Exclusion criteria.** The study excluded female healthcare workers who were expecting, had a hysterectomy or were infertile in the past and were on yearly leave at the time the data were collected.

### 3.4. Sample size determination and sampling techniques

**3.4.1. Sample size determination.** For the first specific objective sample size was determined using single population proportion formula

Where n = minimum sample size required

P = Estimated proportion of utilization of LACM = 0.23(22.7%). based) on research done in [19]

d = margin of error between sample and population (0.05)

Z α/2 = critical value at 95% interval, which is 1.96

$$n = \frac{(1.96)2 * (0.23) * (0.77)}{(0.05)2} = \mathbf{272}$$

For possible non-respondent during the data collection time, 10% were added which give a final sample size of **299**. For the second objective (**Table 1**) we used a double population formula using Epi info7 software for individual factors at 95% confidence level with 5% marginal of error, 80% power and 1:1 ratio of exposed to unexposed.

**3.4.2. Sampling procedure.** There are thirteen hospitals in South Wollo Zone, the study subjects were proportionally allocation (**Fig 1**) from each hospital and randomly selected by using the simple random sampling method. Finally, the study participant was selected from the healthcare providers register card by using lottery method in selected hospitals.

**Table 1. Sample size determination for factor associated with utilization of LACMs service using studies, 2021.**

| Variable | LACM utilization | | AOR | Sample size | Final sample size (10% non-respondent add) | Reference |
|---|---|---|---|---|---|---|
| | Exposed (%) | None exposed (%) | | | | |
| Husband support | 88.8 | 97.3 | 4.62 | 322 | 354 | [19] |
| Family monthly income <5000 | 91.2 | 78.7 | 2.81 | 286 | 314 | [19] |
| Desire of children 0–2 | 62.9 | 37.1 | 0.347 | 132 | 145 | [19] |
| Maximum sample size | | | | 322 | **354** | |

From the above result, the maximum sample size was obtained from the second objective which is **354**.

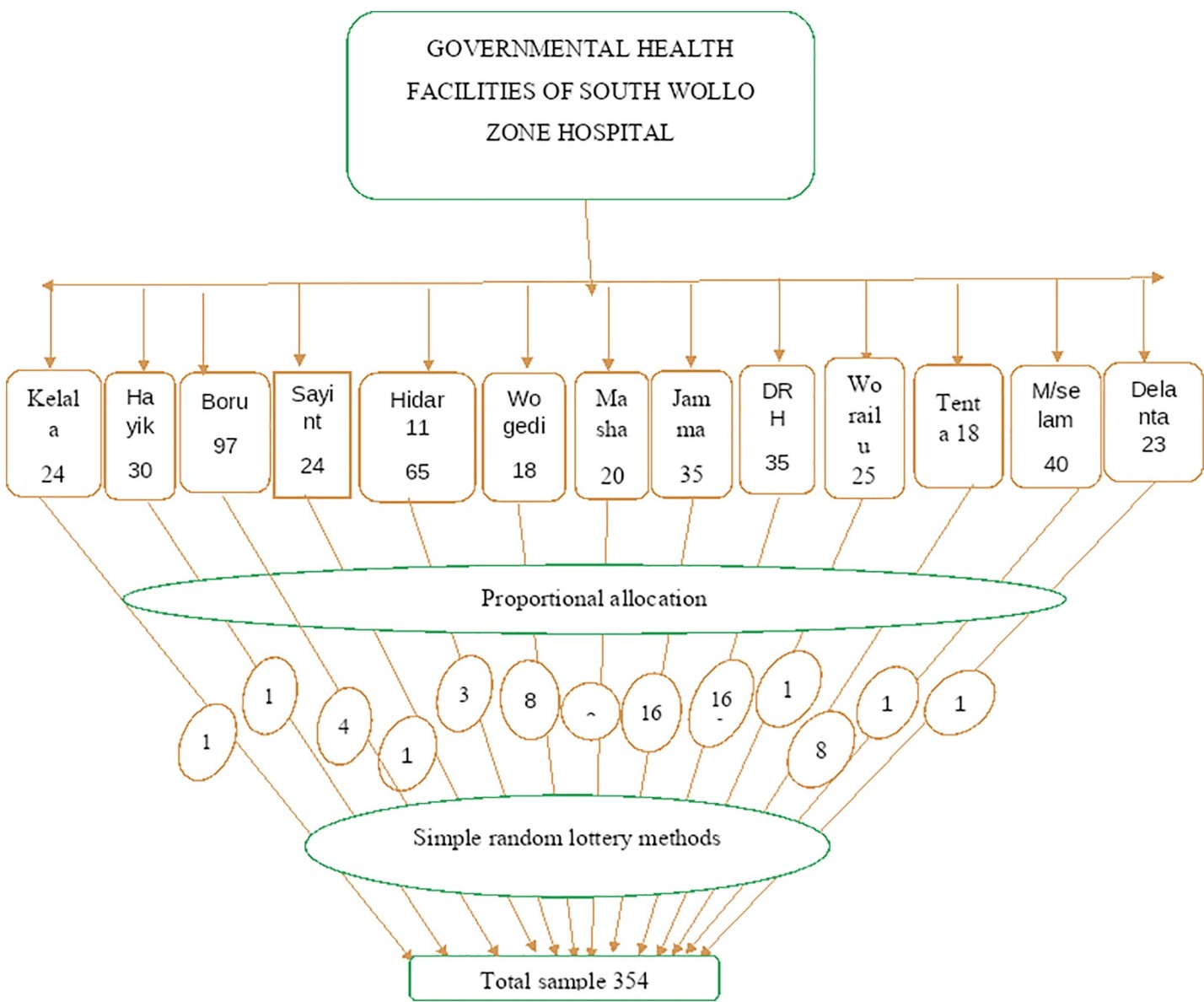

**Fig 1. Schematic presentation of sampling procedure for utilization of LARCs in south Wollo zone hospitals 2021.**

### 3.5. Variable

**3.5.1. Dependent variable.**  Long-acting contraceptive method of Utilization.

**3.5.2. Independent variable.**  **Socio demographic and Socio-economic factor**:—Age, Ethnicity, Current work position/profession, Marital status, Religion, Husband education. Residence, work experience Husband occupation, Monthly family income, being student, educational level.

**Reproductive factor:**—Number of children ever born, Number of live children, Time to have the next baby, Discussion between partner, Abortion, Desire to have children, started sexual intercourse, Fear of fertility

**Individual factor:**—Fear of side effect, Method preference, Miss of perception, Knowledge of LACRCMs, Source of information, Attitude of LACMs, Pervious use of LACMs, Partner support, Method shift/switch.

### 3.6. Data collection tool and procedure

A self-administered structured and pretested questionnaire was used, which was adopted and used to collect using about the study participants. The questionnaire has six information categories includes: Socio demographic, Reproductive history, utilization, Knowledge and Attitude.

Four diploma nurses' data collectors participated in the data collection. Data collectors were trained to have been informed about how to approach the respondents, objectives of study and to keep the privacy of the respondents. During data collection, supervisor was checking the completeness of the questionnaire and receives the collected and completed questionnaire. Respondent were asked written consented and interviewed. Furthermore, on data collection time, when the sampled women were not be accessed for absence, up to two attempts were tried for interviewing to decrease the non-response rate.

### 3.7. Data quality management

To assure the quality of the data, the collected data were checked by the principal investigators for the completeness of the questionnaires and necessary correction was done. The data collectors were given training about the objective, relevance of the study, confidentiality of information, and study participants' rights before actual data collection. The questionnaire was pretested before the actual data collection by taking around 5% of the total calculated sample size to check its consistency, validity and acceptability of the questioners. So, vague checklists and other similar mistakes were corrected before the actual data collection has begun. Regular supervision and follow-up was made by the principal investigator.

### 3.8. Data processing and analysis

Data were entered and coded into the Epi-data 4.1 then checked and cleaned for completeness and consistency. The data was transferred to a Statistical Package for the Social Sciences (SPSS) version 23 for analysis.

Descriptive statistics were summarized in frequency, graph and percentages. Binary logistic regression was conducted and crude odds ratio (COR), with 95% Confidence Interval (CI) was estimated to select the candidate variables for the final model. Then, a variable with a p-value of $< 0.2$ at binary logistic regression was taken into a multi variable logistic regression to control con-founding (multicolinearity was checked). Hosmer-Lemeshow goodness-of-fit with step-wise (enter method) logistic regression was used to test for model fitness. Adjusted odds ratio (AOR) with 95% CI was estimated to assess the presence of association at multi-variable logistic regression. Lastly, variables with a p-value of $< 0.05$ were considered statistically significant predictors of the outcome variable.

## 4. Result

### 4.1. Sociodemographic characteristics

A total of 354 reproductive-age female healthcare workers were included in the analysis making the response rate 100% as shown in **Table 2**. The mean age of the participants was 28 years with SD ±4.6 years. Two hundred thirty-eight (67.2%) participants were married and 230 (65%) hold a degree educational level. One hundred eighty-eight (53.1%) participants were nursing in their profession. Regarding income, 238 (75.7%) earned more than 5000 ETB per month **(Table 2)**.

### 4.2. Reproduction related characteristics of the study participant

Three hundred eleven participants (87.8%) have started sexual intercourse with a mean age of 19.8 (SD ± 2.747) years, and out of them, 290 (93.25%) participants made their first sex before the age of 18 years. Concerning the parity of women, 138 (69.3%) gave at least one birth before. Of the 40 (11.3%) participants had a history of abortion; of these, 33 (82.5%) faced abortion for one time as shown below in **Table 3**.

### 4.3. Attitudes towards the utilization of long-acting contraceptive methods

Among the study participants women 114 (32.2%) agreed that insertion of IUCD doesn't lead to loss of privacy and 157 (44.4%) said that Implant can't interfere with routine activities (**Table 4**). Of the 131 (37%) women thought that insertion and removal of the Implant were not highly painful. And 143 (40.4%) of them reported that the Implant caused irregular

**Table 2. Socio-demographic characteristic for factor associated with utilization of LACMs service in south wollo zone hospitals, 2021.** N = 354.

| Socio demographic variable of respondents | | Frequency (n = 354) | Percentage |
|---|---|---|---|
| **Age group** | 24 or less | 73 | 20.6% |
| | 25 or more | 281 | 79.4% |
| **Educational level** | Diploma | 124 | 35% |
| | Degree and above | 230 | 65% |
| **Profession** | Nurse | 188 | 53.1% |
| | Midwife | 62 | 17.5% |
| | Doctor | 7 | 2% |
| | Laboratory | 50 | 14.1% |
| | HIT | 9 | 2.5% |
| | Anesthesia | 6 | 1.7% |
| **Experience** | 10 or less year | 335 | 94.6% |
| | 10 or more year | 19 | 5.4% |
| **Marital status** | Unmarried | 116 | 32.8% |
| | Married | 238 | 67.2% |
| **Husband educational** | Lower class and secondary school | 20 | 5.6% |
| | Diploma | 49 | 13.8% |
| | Degree and above | 181 | 51.1% |
| **Husband occupational** | Governmental employee | 199 | 56.2% |
| | Self-employee | 38 | 10.7% |
| | Merchant | 13 | 3.7% |
| **Monthly income** | < 5000 | 86 | 24.3% |
| | ≥ 5000 | 268 | 75.7% |
| **Learning now** | Yes | 320 | 90.4% |
| | No | 34 | 9.6% |

**Table 3. Reproductive history of participants in south wollo zone hospitals, 2021, n = 354.**

| Variable | Category | Frequency | Percent (%) |
|---|---|---|---|
| Started sexual intercourse (n = 354) | Yes | 311 | 87.8% |
| | No | 43 | 12.2% |
| Age at first sex (n = 311) | <18 | 290 | 93.25% |
| | ≥ 18 | 21 | 6.75% |
| Age at first marriage(n = 254) | <18 | 25 | 9.85% |
| | ≥ 18 | 229 | 90.15% |
| Ever birth given | Yes | 199 | 65.7% |
| | No | 104 | 34.3% |
| Age at first birth (n = 199) | <18 | 10 | 5% |
| | ≥ 18 | 189 | 95% |
| Parity(n = 199) | 1–2 times | 138 | 69.4% |
| | 3–4 times | 51 | 25.6% |
| | ≥ 5 times | 10 | 5% |
| Number of alive children | 1–2 | 139 | 39.3% |
| | 3–4 | 52 | 14.7% |
| | ≥ 5 | 7 | 2% |
| History of abortion | Yes | 40 | 11.3% |
| | No | 314 | 88.7% |
| Number of abortions | One | 33 | 82.5% |
| | Two and above | 7 | 17.5% |
| Future desire to fertility | Yes | 324 | 91.5 |
| | No | 30 | 8.5% |
| Number of want children in the future | 1–2 children | 47 | 14.5% |
| | 3–4 children | 202 | 62.4% |
| | ≥ 5 children | 75 | 23.1% |
| Responsible for deciding to have children (n = 324) | Wife | 31 | 9.6% |
| | Husband | 18 | 5.55% |
| Joint discussion (both) | | 275 | 84.9% |

**Table 4. Attitude towards long-acting contraceptive among female healthcare providers in South Wollo Zone governmental hospitals, 2021.** n = 354.

| Attitude about LACMs | strongly disagree | disagree | not sure | Agree | strongly agree |
|---|---|---|---|---|---|
| Using an Implant cause Irregular bleeding. | 20 (5.65%) | 66 (18.6%) | 46 (13%) | 143(40.4%) | 79 (22.3%) |
| Insertion and removal of Implant is highly painful | 58 (16.4%) | 131 (37%) | 63 (17.8%) | 84(23.7%) | 18 (5.1%) |
| Implant can't interfere with routine activities. | 29 (8.2%) | 62 (17.5%) | 28 (7.9%) | 157(44.4%) | 78 (22%) |
| Implants do not move through the body after insertion. | 25 (7.1%) | 64 (18.1%) | 64 (18.1%) | 136(38.4%) | 65 (18.4%) |
| IUCD insertion doesn't lead to loss of privacy. | 31 (8.8%) | 40 (11.3%) | 114 (32.2%) | 102 (28.8%) | 67 (18.9%) |
| IUCD doesn't move through the body after insertion. | 19 (5.4%) | 45 (12.7%) | 63 (17.8%) | 155(43.8%) | 72(20.3%) |
| IUCD has no interference with sexual intercourse desire. | 19 (5.4%) | 43(12.1%) | 70(19.8%) | 137(38.7%) | 85(24%) |
| IUCD doesn't restrict normal activities. | 19(5.4%) | 620(5.6%) | 47(13.3%) | 161(45.5%) | 107(30.2%) |
| Long-acting contraceptives should not be used only by women who do not want more children | 24(6.8%) | 79(22.3%) | 44 (12.3%) | 123(34.7%) | 84 (23.7%) |
| Using long-acting contraceptive cause not ectopic pregnancy | 29 (8.2%) | 55(15.5%) | 71(20.1%) | 132(37.3%) | 67 (18.9%) |

**Table 5. Knowledge of long-acting contraceptive methods among female healthcare providers in South Wollo Zone governmental hospitals, 2021 (n = 354).**

| Knowledge statements | Knowledge on LACM | |
|---|---|---|
| | Yes | No |
| Implants are immediately reversible (become pregnant quickly when removed) | 152 (42.9%) | 202 (57.11%) |
| Implants have a side effect | 327 (92.4%) | 27 (7.6%) |
| Implants effectively protect from unwanted pregnancy | 346 (97.7%) | 8(2.3%) |
| Implants can prevent unwanted pregnancies for 3 up to 5 years | 352 (99.4%) | 2(0.6%) |
| Implants require a minor surgical procedure | 340 (96%) | 14 (4%) |
| Implants prevent STI | 17 (4.8%) | 337 (95.2%) |
| Intra Uterine Devices have a side effect | 192 (82.5%) | 62 (17.5%) |
| IUCD effectively protect from unwanted pregnancy | 327 (92.4%) | 27 (7.6%) |
| Intra Uterine Device can prevent pregnancies for 12 years | 351(99.2%) | 3 |
| IUCD is not appropriate for females at high risk of getting STIs | 211 (59.6%) | 143 (40.6%) |
| IUCD is not interference with sexual intercourse or desire | 155 (43.8%) | 199 (56.2%) |
| IUCD is immediately reversible (become Pregnant quickly when removed) | 230 (65%) | 124 (35%) |
| Intrauterine Device is not Cause cancer | 197 (55.6%) | 157 (44.4%) |

bleeding. One hundred eight-five (52.3%) and 169 (47.74%) of the female healthcare providers had positive and negative attitudes towards practicing LACMs respectively.

## 4.4. Knowledge of long-acting contraceptive method of study participant

Respondent's level of knowledge about the long-acting contraceptive methods was assessed by asking fourteen questions (**Table 5**) and then categorized based on the percent of knowledge as "good knowledge" (those who knew the above mean knowledge question), and "poor knowledge" those who know below the mean knowledge questions. One hundred fifty-nine (44.9%) of female healthcare providers were in the category of good knowledge towards LACMs and the remaining one hundred ninety fifty (55.1%) had poor knowledge.

Over half of the female healthcare providers in this study had knowledge both Implants & IUCD 177 (50%) followed by implants 114 (32.2%) and IUCD 63 (17.8%). The majority of the responded to this aware that implant 346 (97.7%) & IUCD (327 (92.4%)) effectively protect from unwanted pregnancy. One hundred fifty-five (43.8%) and 65% of the women were aware that IUCD has no interfere with sexual intercourse or desire and it results in immediate pregnancies after removal of IUCD, respectively.

The majority (99.2%) of female healthcare providers (**Table 6**) had good knowledge that Intra Uterine Devices can prevent pregnancies for 12 years. Over half of study participants 57.11% thought that implant is not immediately reversible and 35% of participants mentioned that an intrauterine device (IUD) can't be immediately removed. Of the 352(99.4%) participants knew the notion that implants can prevent pregnancy for 3–5 years but 2(0.6%) of them did not know.

## 4.5. Utilization of long-acting family planning methods

Utilization of long-acting family planning methods among female healthcare providers was 33.6% at (95% CI 29–39). Of 68 (57.14%) (**Fig 2**) participants used Implants for three years whereas 20 (16.8%) were using intrauterine devices.

Over one-third (66.38%) of the respondents (**Table 6**) were reported wanting to space children thought as a reason for using the long-acting methods. Almost two-third LARC users (82.7%) received services from government health institutions. The main reason cited by the

**Table 6. LAPMs utilization practice in South Wollo zone hospitals, March, 2021, n = 354.**

| Variables | | Frequency | Percent |
|---|---|---|---|
| Ever used any Modern contraceptives (n = 354) | Yes | 283 | 79.94% |
| | No | 71 | 20.1% |
| Where do you get LAPM (n = 283) | Governmental facilities | 234 | 82.68% |
| | Private hospitals/clinics | 38 | 13.42% |
| | Pharmacy | 2 | - |
| | Nongovernmental organization | 8 | 2.8% |
| Current Utilize of LAPMs (n = 354) | Yes | 119 | 33.62% |
| | No | 235 | 66.38% |
| Type of contraceptive currently utilized (n = 119) | Implants | 99 | 83.2% |
| | IUCD | 20 | 16.8% |
| Reason for not using LAPM (n = 235) | Fear of side effect | 97 | 41.3% |
| | fear of infertility | 71 | 30.21% |
| | Miss understanding | 25 | 10.64% |
| | Have no interest to use | 42 | 17.87% |
| Reason for choosing LAPM n = 119 | Have enough child | 13 | 10.9% |
| | Want to space | 79 | 66.38% |
| | Due to the prolonged duration of use | 13 | 10.9% |
| | Safe for health | 14 | 11.76% |
| Discussions with partner/friends on contraceptive n = 354 | Yes | 246 | 69.5% |
| | No | 108 | 30.5% |

respondents for not using LACM was fear of side effects 97 (41.3%), fear of infertility 71 (30.21%), 42 (17.87%) has no interest to use and 25 (10.64%) misunderstanding respectively. The ever use of long-acting contraceptive methods was 283 (79.94%).

## 4.6. Factors associated with long-acting contraceptive use

Both bi-variable and multi-variable logistic regression analysis was done. The result of multi-variable logistic regression shows that (**Table 7**) discussed with partner, method shifting/

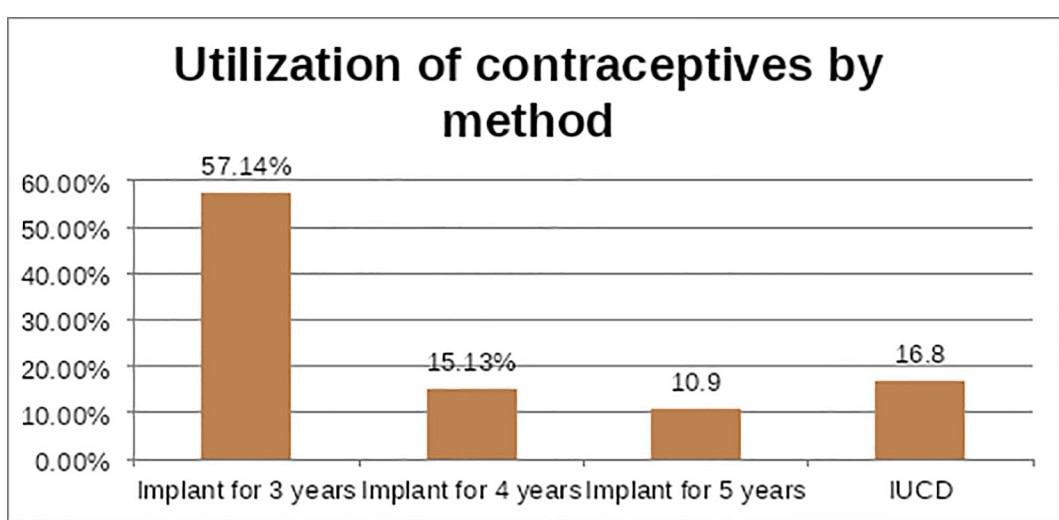

**Fig 2. The utilization contraceptives by method in South wollo zone hospitals, 2021.**

**Table 7. Factors associated with the utilization of long-acting contraceptive methods among reproductive aged female healthcare workers in south wollo zone, Ethiopia, in 2021, n = 354.**

| Explanatory variables | Utilization of LACMs | | COR 95%CI | AOR 95%CI | P-value |
|---|---|---|---|---|---|
| | Yes | No | | | |
| **Age**   <24 | 7 | 66 | 6.249 (2.766–14.115)* | 0.516 (0.39–1.921) | 0.324 |
|    > = 25 | 112 | 169 | 1 | 1 | |
| **Marital status** | | | | | |
|    Unmarried | 8 | 108 | 11.142 (5.507–25.28) * | 2.793 (0.904–8.629) | 0.074 |
|    Married | 111 | 127 | 1 | 1 | |
| **Educational level** | | | | | |
|    Diploma | 35 | 89 | 1.463 (0.910–2.351) * | 0.929 (0.393–2.199) | 0.867 |
|    Degree and above | 84 | 146 | 1 | 1 | |
| Income   <4999 | 19 | 80 | 2.099 (1.191–3.698) * | 0.899 (0.306–2.644) | 0.846 |
|    > = 5000 | 100 | 168 | 1 | 1 | |
| Method shifts   Yes | 98 | 80 | 9.042 (5.253–15.563) * | 4.320 (2.285–8.102) | **0.000**\*\*\* |
|    No | 21 | 155 | 1 | 1 | |
| Ever birth   Yes | 111 | 85 | 16.199 (6.645–39.48)* | 15.670 (5.065–48.49) | **0.000**\*\*\*o |
|    No | 6 | 98 | 1 | 1 | |
| **Want more children in future** | | | | | |
|    Yes | 102 | 222 | 0.351 (0.164–0.751) * | 1.007(0.413–2.457) | 0.987 |
|    No | 17 | 13 | 1 | | |
|    Discuss with partner   Yes | 103 | 207 | 2.355 (1.175–4.722) * | 2.277 (1.026–5.055) | **0.043**\* |
|    No | 16 | 123 | 1 | 1 | |
| Attitude   Negative | 44 | 125 | 0.516 (0.329–0.811) * | 1.613(0.888–2.927) | 0.116 |
|    Positive | 75 | 110 | 1 | 1 | |
| Knowledge   Poor | 59 | 136 | 1.397 (0.897–2.176) * | 1.887 (1.020–3.491) | **0.043**\* |
|    Good | 60 | 99 | 1 | 1 | |

*Statistically significant (p value < 0.05)

***p-value < 0.0001.

switching contraceptive, knowledge, and ever birth was identified as significant determinants of utilization of long-acting contraceptive use among healthcare providers.

Female healthcare providers who discussed with a partner about LACMs were two times more likely to use the method compared to those who did not discussed with their partners. [AOR = 2.277, 95% CI: 1.026–5.055].

Those who shifted or switched long-acting contraceptive methods were four times more likely to utilize long-acting contraceptive methods than those who didn't use long-acting contraceptive methods [AOR = 4.302; 95% CI: 2.285–8.102].

Female healthcare providers who had good knowledge were nearly two times more likely to utilize long-acting contraceptive methods than those who had poor knowledge [AOR = 1.887, 95% CI: 1.020–3.491]. Another significant factor is, those who ever birth are 15 times more likely to use long-acting contraceptive methods compared to those who do not ever birth (AOR = 15.670, 95% CI: 5.065–48.49].

## 5. Discussion

The current study found that the proportion of Long-acting contraceptive utilization among female healthcare providers, which was 33.6%. Discuss with the partners, method shifted/

switched, knowledge of the respondents and ever giving birth were significant associated factors with the utilization of long-acting contraceptive methods. This study prevalence was higher compared to studies conducted in the East Gojjam zone, northern Ethiopia, which is 22.7% [19]. Similarly, this study result is higher than the prevalence reported from mini EDHS 2019 nationally (9% Implants and 2% IUCD (11%) [2]. Comparison of the findings with Ethiopia's FP 2020 commitments confirms that it is lower [18]. However, this study is in line with studies done in, Adaba town West Arsi zone Oromia, Arsi Negele town, Hossana town and Gondar city Administrative, which show that LARCs use was [24–27] A possible explanation for this might be that due to the study design and population differences; this study was institution-based and conducted on government-employed study population, which have highly accessed information and services. But many studies were conducted in the community-based design and in different groups of the population which means having different socioeconomic statuses and educational levels. The other reason might be that comprehensive long-acting reversible contraceptive training and accessibility of services was given to most healthcare providers.

This study showed that respondents who discussed LACMs with their partners were more likely to use LACMs than those who had no discussion. This is supported by the study done in Uganda, East Gojjam Zone, Addis Ababa public health centres, Hossana, and Debre Markos town [19,26,28–30], which showed that discussion with a partner was positively associated with LACMs use. This result may be explained by the fact that Ethiopian wives including female healthcare service providers have values and respect the attitude of their partners. In addition to this, male/partner involvement & support on long-acting contraceptive methods help women to adopt more convenient methods with confidence.

Female healthcare providers who had good knowledge of long-acting contraceptive methods were two times more likely to utilize long-acting contraceptive methods than women who had poor knowledge. This finding was consistent with the study done in Bahir Dar City Public Health Facility, Hossana town, in Chinese healthcare provider, Gondar city [6,26,31–33].

The possible justification might be to those who had good knowledge has ready to be aware and when possible, side effects and infertility happen to them and are able to make a decision on how to solve it by using long-acting contraceptive methods and they can choose the preferred method according to their fertility need.

Female healthcare providers who have ever birth were Fifty times more likely to utilize long-acting contraceptive methods than who had not ever birth. The possible justification for this might be due to; having ever birth made them want to space child, limit the number of children and be safe for health by using long-acting contraceptive methods.

Method switching has a positive association with long-acting contraceptive utilization. This result is in line with to study conducted in the Silti district [34]. This might be due to having experience in using different types of family planning methods helps to understand its importance and have access to counseling services.

In addition to the finding this, National reproductive health strategies state that, build the competency of health workers to provide long-acting and permanent methods of contraception, involving men to support their partners, implement FP quality improvement initiatives and put in place regular supervision to offer compassionate client-friendly service, respecting choice, safety and quality and train all service providers (doctors, nurses, midwives and HEWs) on contraception are some of the strategic interventions to improve utilization of family planning. The limitation of this study was that it does not include all health centers and health post healthcare providers.

## 6. Conclusion

The study shows that utilization of long-acting contraceptive methods among female health-care providers in the South Wollo Zone, public health facilities is low as compared to the national plan. Discussions with their partner towards the utilization of LACMs, the method shifting/switching knowledge towards the utilization of LACMs and ever-birth were statically significant factors with the utilization of LACMs. Strengthening the training for health professionals and teaching the female care provider exhaustively about the LACMs needs discussion with the partners as best to increase the utilization. A further longitudinal study with a large sample size is recommended with factors by adding a qualitative study method.

## Supporting information

**S1 Data.**
(SAV)

## Author Contributions

**Conceptualization:** Aragaw Hamza Yimer, Mehdi Shumiye Seid, Fasil Walelign, Yitayish Damtie.

**Formal analysis:** Mehdi Shumiye Seid.

**Methodology:** Ahmed Muhye Seid.

**Validation:** Fasil Walelign.

**Writing – original draft:** Aragaw Hamza Yimer.

**Writing – review & editing:** Yitayish Damtie, Ahmed Muhye Seid.

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
