## [Decision Letter · Decision Letter 0]

27 Jun 2022

PGPH-D-22-00521

“Utilization of Long –Acting Contraceptive Methods and Associated Factor among Female Health Care Providers in South Wollo Zone Hospitals, North East, Ethiopia. A cross-sectional multicenter study”

Dear Yimer,

Thank you for submitting your manuscript to PLOS Global Public Health. After careful consideration, we feel that it has merit but does not fully meet PLOS Global Public Health’s publication criteria as it currently stands. Therefore, we invite you to submit a revised version of the manuscript that addresses the points raised during the review process.

We look forward to receiving your revised manuscript.

Kind regards,

Collins Otieno Asweto, PhD

Academic Editor

Journal Requirements:

1. We suggest you thoroughly copyedit your manuscript for language usage, spelling, and grammar. If you do not know anyone who can help you do this, you may wish to consider employing a professional scientific editing service.

b. State what role the funders took in the study. If the funders had no role in your study, please state: “The funders had no role in study design, data collection and analysis, decision to publish, or preparation of the manuscript.

3. Please update the Funding Information section in the system and ensure that it matches with Financial Disclosure Statement.

4. In the online submission form, you indicated that "The data will be available at request". All PLOS journals now require all data underlying the findings described in their manuscript to be freely available to other researchers, either 1. In a public repository, 2. Within the manuscript itself, or 3. Uploaded as supplementary information.

5. Please ensure that the Title in your manuscript file and the Title provided in your online submission form are the same.

6. Please provide separate figure files in .tif or .eps format and removed from the manuscript file.

Reviewers' comments:

Reviewer's Responses to Questions

**Comments to the Author**

1. Does this manuscript meet PLOS Global Public Health’s publication criteria? Is the manuscript technically sound, and do the data support the conclusions? The manuscript must describe methodologically and ethically rigorous research with conclusions that are appropriately drawn based on the data presented.

Reviewer #1: Partly

Reviewer #2: Partly

2. Has the statistical analysis been performed appropriately and rigorously?

Reviewer #1: No

Reviewer #2: Yes

3. Have the authors made all data underlying the findings in their manuscript fully available (please refer to the Data Availability Statement at the start of the manuscript PDF file)?

Reviewer #1: No

Reviewer #2: Yes

4. Is the manuscript presented in an intelligible fashion and written in standard English?

Reviewer #1: No

Reviewer #2: No

5. Review Comments to the Author

Reviewer #1: Abstract

Background:

First line: where was the utilization low? Incomplete sentence

Second line says no study while third line mention about the study: confuses the reader

Method: need re-writing on the analysis part to have better clarity. For instance Bivariable and multivariable logistic regression analyses were also performed- for what?

Result: The result is confusing- particularly the interpretation of OR.

Background

Lengthy explanations of technical issues, global issues- which could be shortened.

Suggest adding up more context of Ethiopia.

The statement of problem is kept separately which is not inline with the journal format. The statement should explain briefly about why female health care providers?

Methods

How many hospitals were selected? How were they selected?

Sampling process is confusing. The author has calculated sample size as per objective assuming two outcomes (associations), which is fair but not sure how 354, was decided.

Was ethical approval obtained?

Result

The # of descriptive tables should be reduced and kept it as a supporting tables.

Need to re-check the statistical data. For instance the interpretation says- Female health care providers who had good knowledge were nearly two times more likely to

utilize long-acting contraceptive methods than those who had poor knowledge [AOR=1.887, 95% CI: 1.020-3.491]- while its exactly opposite at the table. Some significant associations has very high CI intervals. For instance (CI: 5.065- 48.49) for ever-birth variable.

Discussion

Suggest comparing the data with the national scenario or other women groups in the country

Add up the strengths and expand the limitations

Add up recommendations to scale up the LACM uptake

Finally suggest- major writing editing interms of grammars, clarity and consistency

Reviewer #2: 1. The article bears a lot of resemblance to another published article with a similar topic but which took place in North West Ethiopia https://doi.org/10.1155/2019/5850629, reference 19 in this article. The first few pages bear similar flow, I will expect the authors to do more on presenting the background with peculiar information from the region of study. As this seems to be a replicate study, the justification for conducting the study is not strong enough. It is not clear what extra information does the study aims to add to the body of knowledge that was not available in similar studies?

2. The presentation of the statistics should be improved, for example: proportion of the Health workers using LAFP is 33.6% was not shown in Figure 2, it is only available in Table 6. This should be clearly stated.

It is not clearly shown how and the multi-variable logistic regression that was done in the analysis.

4. This article will benefit from intense copy editing and correction of the typos and ensure correct use of English language and clarity of the message in the different sections of the manuscript.

6. PLOS authors have the option to publish the peer review history of their article (what does this mean?). If published, this will include your full peer review and any attached files.

**Do you want your identity to be public for this peer review?** For information about this choice, including consent withdrawal, please see our Privacy Policy.

Reviewer #1: No

Reviewer #2: No

---

## [Decision Letter · Decision Letter 1]

2 Nov 2022

PGPH-D-22-00521R1

“Utilization of long–acting contraceptive methods and associated factors among female health-care providers in South Wollo Zone hospitals, Northeast, Ethiopia. A cross sectional multicenter study”.

Dear Yimer

Thank you for submitting your manuscript to PLOS Global Public Health. After careful consideration, we feel that it has merit but does not fully meet PLOS Global Public Health’s publication criteria as it currently stands. Therefore, we invite you to submit a revised version of the manuscript that addresses the points raised during the review process.

EDITOR: Note that your manuscript require English editing.

We look forward to receiving your revised manuscript.

Kind regards,

Collins Otieno Asweto, PhD

Academic Editor

Journal Requirements:

b. If any authors received a salary from any of your funders, please state which authors and which funders.

3. Your current Financial Disclosure states, “Wollo University supported to conduct this study. The funders had no role in study design, data collection and analysis, decision to publish, or preparation of the manuscript”. However, your funding information on the submission form indicatesthat you did not receive funding. Please indicate by return email the full and correct funding information for your study and confirm the order in which funding contributions should appear. Please be sure to indicate whether the funders played any role in the study design, data collection and analysis, decision to publish, or preparation of the manuscript.

4. In the online submission form, you indicated that "The data cannot be made publicly available for ethical or legal reasons. It will be available at request. The data will be available at request". All PLOS journals now require all data underlying the findings described in their manuscript to be freely available to other researchers, either 1. In a public repository, 2. Within the manuscript itself, or 3. Uploaded as supplementary information.

Additional Editor Comments (if provided):

Reviewers' comments:

Reviewer's Responses to Questions

**Comments to the Author**

1. If the authors have adequately addressed your comments raised in a previous round of review and you feel that this manuscript is now acceptable for publication, you may indicate that here to bypass the “Comments to the Author” section, enter your conflict of interest statement in the “Confidential to Editor” section, and submit your "Accept" recommendation.

Reviewer #2: All comments have been addressed

Reviewer #3: All comments have been addressed

2. Does this manuscript meet PLOS Global Public Health’s publication criteria? Is the manuscript technically sound, and do the data support the conclusions? The manuscript must describe methodologically and ethically rigorous research with conclusions that are appropriately drawn based on the data presented.

Reviewer #2: Yes

Reviewer #3: Yes

3. Has the statistical analysis been performed appropriately and rigorously?

Reviewer #2: Yes

Reviewer #3: No

4. Have the authors made all data underlying the findings in their manuscript fully available (please refer to the Data Availability Statement at the start of the manuscript PDF file)?

Reviewer #2: No

Reviewer #3: Yes

5. Is the manuscript presented in an intelligible fashion and written in standard English?

Reviewer #2: Yes

Reviewer #3: No

6. Review Comments to the Author

Reviewer #2: Most of the comments have been taken care of and done well. The new additions in yellow seems to be added after the copy editing has been done, kindly take through same copy editing.

Reviewer #3: Reviewer´s comments

Kindly edit the English. This paper needs to be edited by an English native speaker.

“ COR, CI, AOR”. Kindly write these in full and put abbreviations in brackets the first time you write about them. You can use abbreviations thereafter.

You used logistic regression analysis, but you don´t state anywhere whether you tested for multicollinearity. Did you check for multicollinearity?

In table 3, you indicated that 43 out of 354 of the study participants have never “started sexual intercourse”, meaning they are virgins. You included the 43 virgins in your “LAPMS utilization practice analysis”. Why would a virgin use LAPM? LAPMs prevent pregnancy, why would a sexually inactive person use them? Kindly motivate this.

7. PLOS authors have the option to publish the peer review history of their article (what does this mean?). If published, this will include your full peer review and any attached files.

**Do you want your identity to be public for this peer review?** For information about this choice, including consent withdrawal, please see our Privacy Policy.

Reviewer #2: No

Reviewer #3: **Yes: **Mary Luwedde

---

## [Decision Letter · Decision Letter 2]

20 Jan 2023

PGPH-D-22-00521R2

“Utilization of long–acting contraceptive methods and associated factors among female health-care providers in South Wollo Zone hospitals, Northeast, Ethiopia. A cross sectional multicenter study”.

Dear Dr. Aragaw Yimer,

Thank you for submitting your manuscript to PLOS Global Public Health. After careful consideration, we feel that it has merit but does not fully meet PLOS Global Public Health’s publication criteria as it currently stands. Therefore, we invite you to submit a revised version of the manuscript that addresses the points raised during the review process.

The main issue is that the manuscript contains spelling and grammatical errors, which impede clarity. We kindly request you thoroughly copyedit your manuscript for language usage, spelling, and grammar. If you do

not know anyone who can help you do this, you may wish to consider employing a professional scientific

editing service. 

We look forward to receiving your revised manuscript.

Kind regards,

Katrien Janin

Staff Editor

Journal Requirements:

2. Please ensure that the funders and grant numbers match between the Financial Disclosure field and the Funding Information tab in your submission form. Note that the funders must be provided in the same order in both places as well.

Additional Editor Comments (if provided):

Reviewers' comments:

Reviewer's Responses to Questions

**Comments to the Author**

1. If the authors have adequately addressed your comments raised in a previous round of review and you feel that this manuscript is now acceptable for publication, you may indicate that here to bypass the “Comments to the Author” section, enter your conflict of interest statement in the “Confidential to Editor” section, and submit your "Accept" recommendation.

Reviewer #2: (No Response)

2. Does this manuscript meet PLOS Global Public Health’s publication criteria? Is the manuscript technically sound, and do the data support the conclusions? The manuscript must describe methodologically and ethically rigorous research with conclusions that are appropriately drawn based on the data presented.

Reviewer #2: Yes

3. Has the statistical analysis been performed appropriately and rigorously?

Reviewer #2: Yes

4. Have the authors made all data underlying the findings in their manuscript fully available (please refer to the Data Availability Statement at the start of the manuscript PDF file)?

Reviewer #2: Yes

5. Is the manuscript presented in an intelligible fashion and written in standard English?

Reviewer #2: No

6. Review Comments to the Author

Reviewer #2: The major deficit of this manuscript is the English. I have consistently requested that the authors get a competent English copy editor versed in scientific writing to review the submission. It is very hard reading the texts with many errors in language. I noted that the major lapses are from page 10 to the end. The key information is there, but the use of English is poor. It has improved from the first submission, but not at a stage where this should be published.

7. PLOS authors have the option to publish the peer review history of their article (what does this mean?). If published, this will include your full peer review and any attached files.

**Do you want your identity to be public for this peer review?** For information about this choice, including consent withdrawal, please see our Privacy Policy.

Reviewer #2: No

---

## [Editor Report · Decision Letter 3]

15 Feb 2023

“Utilization of long–acting contraceptive methods and associated factors among female health-care providers in South Wollo Zone hospitals, Northeast, Ethiopia. A cross sectional multicenter study”.

PGPH-D-22-00521R3

Dear Mr Yimer,

We are pleased to inform you that your manuscript '“Utilization of long–acting contraceptive methods and associated factors among female health-care providers in South Wollo Zone hospitals, Northeast, Ethiopia. A cross sectional multicenter study”.' has been provisionally accepted for publication in PLOS Global Public Health.

Best regards,

Julia Robinson

Executive Editor